# The Hearing Impairment Ontology: A Tool for Unifying Hearing Impairment Knowledge to Enhance Collaborative Research

**DOI:** 10.3390/genes10120960

**Published:** 2019-11-21

**Authors:** Jade Hotchkiss, Noluthando Manyisa, Samuel Mawuli Adadey, Oluwafemi Gabriel Oluwole, Edmond Wonkam, Khuthala Mnika, Abdoulaye Yalcouye, Victoria Nembaware, Melissa Haendel, Nicole Vasilevsky, Nicola J. Mulder, Simon Jupp, Ambroise Wonkam, Gaston K. Mazandu

**Affiliations:** 1Division of Human Genetics, Department of Pathology, University of Cape Town, Health Sciences Campus, Anzio Rd, Observatory 7925, South Africa; giant.plankton@gmail.com (J.H.); mnynol006@myuct.ac.za (N.M.); smadadey@st.ug.edu.gh (S.M.A.); oluwafemi.oluwole@uct.ac.za (O.G.O.); wonkamedmond@yahoo.fr (E.W.); mnikakhuthala@gmail.com (K.M.); YLCABD001@myuct.ac.za (A.Y.); vnembaware@gmail.com (V.N.); 2Department of Neurology, Point G Teaching Hospital, University of Sciences, Techniques and Technology, Bamako, Mali; 3West African Centre for Cell Biology of Infectious Pathogens, College of Basic and Applied Sciences, University of Ghana, Accra, Ghana; 4School of Medicine, Oregon Health & Science University, Portland, OR 97217, USA; haendel@ohsu.edu (M.H.); vasilevs@ohsu.edu (N.V.); 5Computational Biology Division, Department of Integrative Biomedical Sciences, University of Cape Town, Health Sciences Campus. Anzio Rd, Observatory 7925, South Africa; nicola.mulder@uct.ac.za; 6European Bioinformatics Institute (EMBL-EBI), Cambridge CB10 1SD, UK; simon.jupp@gmail.com; 7African Institute for Mathematical Sciences, 5-7 Melrose Road, Muizenberg 7945, Cape Town, South Africa

**Keywords:** hearing impairment, hearing loss, ontology, data harmonization, meta-analysis

## Abstract

Hearing impairment (HI) is a common sensory disorder that is defined as the partial or complete inability to detect sound in one or both ears. This diverse pathology is associated with a myriad of phenotypic expressions and can be non-syndromic or syndromic. HI can be caused by various genetic, environmental, and/or unknown factors. Some ontologies capture some HI forms, phenotypes, and syndromes, but there is no comprehensive knowledge portal which includes aspects specific to the HI disease state. This hampers inter-study comparability, integration, and interoperability within and across disciplines. This work describes the HI Ontology (HIO) that was developed based on the Sickle Cell Disease Ontology (SCDO) model. This is a collaboratively developed resource built around the ‘Hearing Impairment’ concept by a group of experts in different aspects of HI and ontologies. HIO is the first comprehensive, standardized, hierarchical, and logical representation of existing HI knowledge. HIO allows researchers and clinicians alike to readily access standardized HI-related knowledge in a single location and promotes collaborations and HI information sharing, including epidemiological, socio-environmental, biomedical, genetic, and phenotypic information. Furthermore, this ontology illustrates the adaptability of the SCDO framework for use in developing a disease-specific ontology.

## 1. Introduction

Hearing impairment (HI), the partial or total inability to hear, is a communication barrier and language development impediment. It can thus have a huge effect on one’s quality of life [1]. HI has the highest rate for age-standardized disability of life in the world [2,3]. According to the most recent World Health Organization (WHO) estimates [4], over six percent of the world’s population, representing approximately 460 million individuals, are currently living with a disabling HI, of which 93% are adults and mostly males (242 million males vs 190 million females). The financial burden associated with HI, which includes costs for healthcare, education, social support, and loss of productivity [5], is estimated to be 750 billion US dollars annually [6]. Even though 60% of HI cases can be prevented [5], the number of cases is expected to significantly increase to over 900 million in 2050 [4] with huge negative economic implications, unless action is taken. As a matter of urgency, there is a need to strengthen collaborative HI research efforts aimed at curbing the projected increased burden of HI globally. The Global Hearing Loss project (https://thespindle.org/project/global-hearing-loss-database/) has highlighted that HI research data is commonly unstructured, stored in natural language format, and hardly shared. The general lack of standardization of research data on rare or neglected diseases across studies [7] hampers presentation, sharing, integration, and interoperability of important information, such as prevalence, socio-environmental, biomedical, and phenotypic information. The need for harmonized HI datasets motivated the World-Wide Hearing group to develop a standard platform, the Global Hearing Loss Database (GHLD). The GHLD is based on WHO protocols with a web portal and a smartphone application to ease HI data collection and sharing processes. However, given the complexity of HI etiologies and phenotypes, analyses of these datasets and inter-study comparability would require a standard knowledge representation of the HI knowledge domain [8]. A standard knowledge representation of the HI concepts or terms would include concise descriptions to ensure a common understanding of the domain and to enable automated reasoning and inferencing. Moreso, with the constant evolution of biomedical knowledge [7], a human- and machine-readable upgradeable system is needed for standardized and well-defined HI knowledge representation to enhance collaborative research in the field.

Recent advances in artificial intelligence have fostered the use of ontology models to represent knowledge and information-based systems [9] in a human- and machine-readable format to help process, reuse, and re-apply knowledge [10,11]. An ontology is useful in establishing a common and controlled vocabulary system, describing key concepts, properties, and hierarchical relationships between concepts [12], with precise definitions for clear and unambiguous communication. In the biomedical research context, several human disease-related ontologies have been introduced, including the Human Disease Ontology (DO), which consistently defines various concepts encountered in disease domains [13], the Mondo Disease Ontology, which provides a merged and comprehensive cross-species disease ontology (https://monarch-initiative.github.io/mondo/), and the Human Phenotype Ontology (HPO), providing controlled vocabularies of abnormal phenotypes encountered in human disease [14,15]. However, as previously argued in support of developing the Sickle Cell Disease Ontology (SCDO) [5], none of the existing ontologies comprehensively captures related concepts specific to HI due to the complex nature of HI etiologies and phenotypes.

We present the Hearing Impairment Ontology (HIO), built upon the SCDO framework. The HIO was compiled by a working group, which includes HI and ontology experts, who defined, in detail, essential aspects of the HI knowledge domain (e.g., phenotypes, genetics, therapeutics, diagnostics, etc.) and how these aspects are related. Similar to how the SCDO was built around the central concept, ‘Hemoglobinopathy’ [7], the HIO is built around the ‘Hearing Impairment’ concept. However, because HI can be associated with a myriad of phenotypes and/or syndromes, caused by various genetic, environmental, and/or unknown factors, the SCDO framework was adapted to account for the additional complexities of HI. To date, this developing HIO represents the most comprehensive standardized HI domain knowledge portal, which will allow for the application of ontology-driven mining approaches for the identification of pertinent research questions. 

## 2. Materials and Methods

For this first version of the HIO, the working group consisted of experts from an existing Hearing Impairment Genetics Studies in Africa (HI-GENES Africa) project at the University of Cape Town, Division of Human Genetics in the Department of Pathology, and expert ontology developers to provide technical guidance. The experts from the HI-GENES Africa project included PhD students and Postdocs, with a varied range of expertise which included clinicians, biomedical scientists, geneticists, and bioinformatics experts. SCDO curators and developers led the design, following the published ontology development reporting guidelines [16] and best practice.

### 2.1. SCDO Model-Based HIO Development

After attempting to “fit” different forms of HI and their associated causes into the same upper classes used in the SCDO (except replacing ‘Hemoglobinopathy’ with ‘Hearing Impairment’), it was apparent that the HIO model needed to be carefully adjusted to account for the marked differences in the causes and pathophysiology between these diseases. A schema for the HIO was drawn up to formalize how the HIO would be modelled (what main classes would be needed and what relationships would be described between these classes). It is worth noting that these two diseases have complex phenotypic expressions, influenced by several genetic and environmental factors. Since SCDO was built around the central concept ‘Hemoglobinopathy’ to include more factors influencing its phenotypes [7], likewise, HIO is built around the central concept ‘Hearing Impairment’ to ensure that all aspects influencing the disease outcome and phenotypic manifestations are captured. This is achieved by relating different HI concepts specified in the ontology to various factors, including genetic and environmental factors, that contribute to the disease outcome. The overview of different steps in the modeling, from populating the ontology, checking different concepts and relations to the release of the HIO by curators, domain and ontology experts via internal and external reviews, is described in Figure 1.

### 2.2. Building Different HIO Objects

Annotation properties (both required and optional) were re-used from the SCDO. The additional annotation property, ‘deprecated synonym’, was included by the working group in order to indicate when a term has a synonym that is no longer acceptable. Terms to be included in the ontology were added and annotated in a shared online spreadsheet by the working group. The relationships between classes were also captured in shared online spreadsheets (each sheet dedicated to the relationships made by a certain object property). To keep track of terms reused from existing ontologies, the ‘existence in other ontologies’ annotation property was used to assign an ‘existence status’ to each term. The frequency of existence statuses was subsequently used also to evaluate ontology terms unique to the HIO and its contribution to updating HI terms in other ontologies.

### 2.3. Distributed Model-Based HIO Design

Coordinating an ontology development with groups of contributors from heterogeneous specialized backgrounds to derive a unified domain conceptualization is challenging. To ease the process, the online collaborative ontology development tool, WebProtege [17], was used to draft the skeleton structure (labels of terms only) of the HIO in Ontology Web Language (OWL) format. This tool provides a highly distributed ontology content management system, enabling domain experts, ontology curators, and developers to share and update information, and easily visualise the ontology classes and structure.

### 2.4. HIO File Refinement and Evolution

For quality control assurance of different concepts in the ontology and considering the dynamic evolution of the ontology structure, an iterative and collaborative process was used for refining different definitions and properties, as well as the topological structure. These concepts were validated by experts before being included into the hio-edit.owl file in WebProtege. Each term had to be checked by at least two members of the working group, including at least one Hl expert. Once terms were validated during the general online discussion meeting (or internal review), curators added their annotations from the spreadsheet into the WebProtege project (recording in the spreadsheet which terms had been added). Thereafter, ROBOT, an open source tool for automating ontology development workflows and tasks [18], was used to compile the complete ontology release files, which are Web Ontology Language (OWL) and Open Biomedical Ontology (OBO) formats.

## 3. Results

### 3.1. HIO General Description

Other than the ‘HIO’ and ‘deprecated terms’ classes, the HIO currently consists of 10 upper classes (see Figure 2). It contains 495 terms that are topologically connected by 543 links (is_a relationships) and expected to be associated with each other by 45 object properties (excluding the owl:topObjectProperty). Figure 2 shows how the SCDO upper classes were adapted for the HIO. Figure 3 shows the number of concepts in the HIO’s upper level classes (Graph A) and the number of relationships currently asserted between concepts via object properties (Graph B) in this first release version of the ontology.

The following upper classes from the SCDO are included in this first draft of the HIO: Diagnostics, Gene Product, Genetic Phenomena, Mode of Inheritance, Personal Attribute, Phenotype and Therapeutics. A new HIO identifier is assigned to each reused concept from other ontologies and attached to cross references to the source ontology. The SCDO’s central ‘Hemoglobinopathy’ class has been replaced by a new central ‘Hearing Impairment’ class, which contains four main subclasses: ‘Hearing Impairment by Cause’, ‘Hearing Impairment by Ear Affected’, ‘Hearing Impairment by Onset’, and ‘Hearing Impairment by Physiopathology Mechanism’ (see Figure 4), which are comprehensively populated with the current HI domain knowledge, capturing various aspects associated with HI.

Two new (compared to the SCDO) upper classes have been included, namely: ‘Disease Attribute’ and ‘Modifier’ (see Figure 2). The ‘Disease Attribute’ class (see Figure 5) incorporates content similar to the SCDO’s ‘Association’ class but notably also includes a ‘Disease Cause’ sub-class, which was found necessary due to the varied and often complex causes of hearing impairments [19,20,21,22]. The ‘Disease Cause’ class contains the term ‘Unknown Etiology’ and the two subclasses ‘Environmental Disease Cause’ and ‘Intrinsic Disease Cause’ [23,24,25], which are populated comprehensively with factors that cause or contribute in some way to HI.

The ‘Modifier’ class includes the SCDO’s ‘Disease Modifier’ upper class as a sub-class, along with a new ‘Disease Cause Modifier’ class. This additional type of modifier was included because causes of hearing impairment sometimes have modifying factors that determine whether or not the disease is in fact caused/present, e.g., ototoxicity induced by drugs (one cause of hearing impairment) has numerous modifying factors (e.g., dose, duration of therapy, concurrent renal failure, infusion rate, lifetime dose, coadministration with other drugs having ototoxic potential, genetic susceptibility) [26].

The HIO’s central ‘Hearing Impairment’ class links, either directly or indirectly, to sub-classes of all other upper classes through numerous associations, as can be seen in Figure 6. For simplicity, associations made with terms in the ‘Modifier’ upper class are not shown (see these in Figure 7) and only associations with the ‘Disease Cause’ sub-class of the ‘Disease Attribute’ class are displayed. As shown by the large yellow c-shape in Figure 6, the ‘Intrinsic Disease Cause’ class encompasses portions of most of the other upper classes, namely ‘Genetic Phenomena’, ‘Phenotype’, ‘Gene Product’, ‘Personal Attribute’ and ‘Therapeutics’. 

The remaining associations made in the ontology, i.e., with the other disease attributes not in the ‘Disease Cause’ class (namely: ‘Age of Onset’, ‘Mode of Onset’, and ‘Disease Qualifier’ (which includes qualifiers such as ‘Rare and ‘Acquired’)) and with the ‘Modifier’ class, are shown in Figure 7.

### 3.2. Assessing the Relevance of HIO

Of the 493 terms in the ontology (excluding ‘deprecated terms’ and upper ‘owl:Thing’), at the time of writing this, 399 terms have descriptions and are considered as having all minimum required annotations (i.e., *rdfs:label*; *dc:description*; *definition source*, if relevant; *database cross reference*, if applicable; *dc:creator*, if the term was defined by an HIO curator using a source other than available ontologies; *existence in other ontologies*, used to record the existence status of the term prior to inclusion in the HIO; and *hasExactSynonyms,* to indicate synonyms, where relevant). Analysis of the existence statuses ascribed to these terms using the ‘existence in other ontologies’ annotation property (see Table 1) shows that the HIO, even in its first draft, is making contributions towards including and standardizing HI terms that were previously not included in other ontologies. Some of these terms are provided in Table 2 for illlustration. Where applicable, terms unique to the HIO will be recommended for inclusion into other related ontologies, such as the HPO, Orphanet, DO, MESH, and NCIT.

### 3.3. HIO Release and License

The HIO is released every two months with possible special releases when there are significant incidental changes. It is freely available under the Creative Commons Attribution 4.0 Unported License (CC:https://creativecommons.org/licenses/by/4.0/legalcode) and further copyrighted to maintain the quality and integrity of the term vocabulary, meaning that any modification to the HIO can only be done by HIO developers and curators.

### 3.4. Different HIO Access Platforms

In order to foster the dissemination of and easy access to this novel ontology the latest OWL file produced has been uploaded to the NCBO BioPortal at https://bioportal.bioontology.org/ontologies/HIO. This also facilitates the searching and viewing of different HIO concepts. In addition, the OWL and OBO files can be accessible via the GitHub repository at https://github.com/hiodev/hi-ontology.

## 4. Discussion

Making use of ontological reasoning approaches may play a significant role in solving scalability and interoperability issues associated with current large-scale biological high-throughput datasets. This implies that building and maintaining biomedical ontologies is essential, especially in this current data rich era with an extensive consideration of big data analytics. With the contribution of HI domain experts, we have designed the HIO, which enables knowledge acquisition and harmonization, verification and validation of data available in different databases. This ontology is set to be the most comprehensive standardized HI domain knowledge portal, which will allow for the application of ontology-driven mining approaches for the identification of pertinent research questions. The HIO will foster clear and unambiguous communication and also facilitate sharing of information within the field.

### 4.1. HIO Structure, Other Disease Ontologies and HI Online Datasets

As pointed out previously, the HIO reuses concepts from other ontologies, including HPO, DO, and especially SCDO (see Table 1: summarizing the number of HI specific concepts vs reused concepts). These concepts were adjusted, where applicable, to incorporate new concepts specific to HI and relevant in various areas, such as HI subtypes, phenotypic expressions, genetic phenomena and different modes of inheritance. It is worth noting that, although we did not foresee all the adaptations that would be required, the use of the SCDO as a template for the HIO has been a very useful exercise. Whereas the general structure of the SCDO is more readily transferable to other monogenic diseases, we believe the HIO can be used as a template for designing disease-specific ontologies for diseases with a broader range of causes. Finally, note that there exist several online resources storing HI datasets and containing or enabling the retrieval of HI information. Table 3 lists some of these resources.

### 4.2. HIO Potential Future Applications

Even though we have shown the relevance of this new ontology by looking at how many of the classes are HI specific by querying against NCBO BioPortal, it should be noted that an ontology should be applied in order to appropriately assess its impact and suitability. We plan to use this ontology in data representation, which includes data harmonization, interoperability, and integration. For this, the HIO will be an essential resource in designing an ontology-based case report forms, providing essential data elements and controlled terminology. Different datasets can then be mapped to these data elements, making these datasets interoperable, thus easing the data integration process and meta-analysis. In the context of HI research, this will orient data analysis and enable the use of machine learning approaches with sufficient statistical power [31] for predicting disease clinical outcomes [32] and optimal therapeutic interventions, based on the disease pathophysiology mechanisms and other clinical parameters in patient records. It is expected that this HIO will contribute to fostering the subsequent HI research translation into healthcare, inferring knowledge based on patient clinical records, and the development of ontology-powered artificial intelligence medical tools helping in therapeutic interventions, prognosis, and diagnosis, as well as predictive models for an improved understanding of disease processes. 

### 4.3. Challenges and Future Direction

Although the HIO has been designed in a manner that takes into account the various complexities of HI, there is admittedly still much content to be included, both with regards to terms and associations. New discoveries are also regularly being made in this field towards technological advances for diagnostics and therapeutic. This suggests that the ontology should be dynamic, in continuous evolution, keeping the HI knowledge up to date as new knowledge is accessible. There will thus be a need for ongoing input and maintenance of the HIO. For this, there is already a dedicated curation team aiming at assuring the quality and accuracy of the information contained in this ontology and also keeping it updated as HI knowledge evolves.

Going forward, these remaining upper classes of the SCDO will also be evaluated, and where necessary, adapted, for inclusion into the HIO: ‘Research’, ‘Guidelines’, and ‘Quality of Life and Care’. We also plan to use competency questions defined by HI experts to evaluate the scope and domain coverage of the HIO. Beyond the use of competency questions, this ontology will also be assessed on its HI concept inclusion power, i.e., in terms of percentage of HI clinical terminologies from a given database, such as the GHLD database, or HI associated clinical reports or selected literature found in the HIO. This is particularly useful as it will provide an indication on the HIO ability in HI text mining tasks. Finally, the next critical challenge is to introduce this HIO into the dynamic clinical setting. This necessitates the development of testable and actionable health informatics applications to ensure clinical system-wide adherence. As indicated previously, HIO addresses the issue of unifying research clinical data from diverse sources. This ontology already paves the way towards the integration of clinical data into electronic medical records, which should facilitate the development of effective health informatics tools to potentially assist in the public and clinical management of hearing impairment conditions.

## 5. Conclusions

We have developed the HIO, a common controlled HI vocabulary, which is expected to enhance collaborative research. This ontology is currently the most comprehensive and standardized human- and machine-readable resource that unambiguously defines HI concepts and terminology for researchers, patients, and clinicians in order to help process, reuse, and re-apply existing HI knowledge in biomedical research and health-care systems. In the context of big data analytics, this ontology may facilitate retrospective data harmonization and contribute to mapping HI datasets to functional knowledge to enable the subsequent HI research translation into clinical applications and policy guidelines. The HIO will allow researchers, clinicians, and patients to readily access standardized HI-related knowledge in a single location and promote HI data integration, interoperability, and sharing, including epidemiological, socio-environmental, biomedical, genetic, and phenotypic datasets.

## Figures and Tables

**Figure 1 genes-10-00960-f001:**
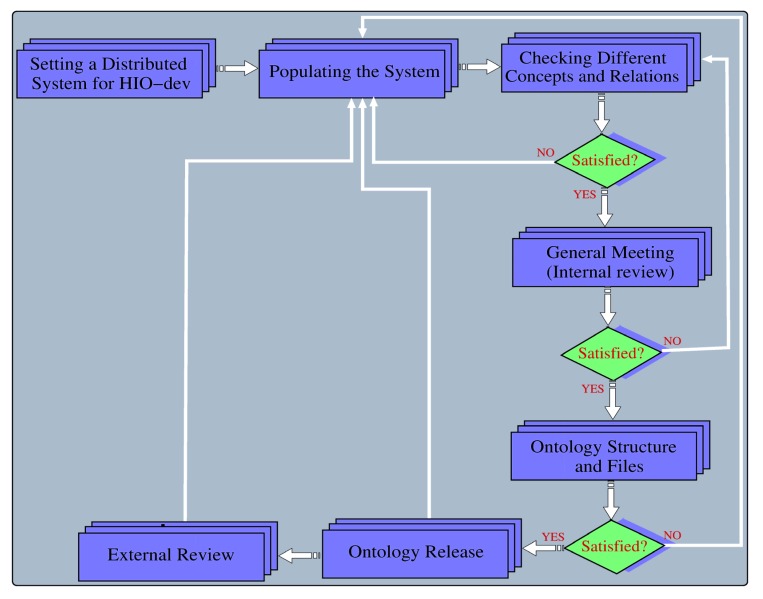
Flow chart of the dynamic and iterative ontology development process. It starts by setting up an online collaborative ontology development tool, WebProtege, which provides a highly distributed ontology content management system, enabling domain experts, ontology curators, and developers to share and update information, and easily visualise the ontology classes and structure. A general discussion meeting (or internal review) is called to share a common understanding of existing Hearing Impairment (HI) knowledge currently included in the ontology and resolve any disagreement about a given concept.

**Figure 2 genes-10-00960-f002:**
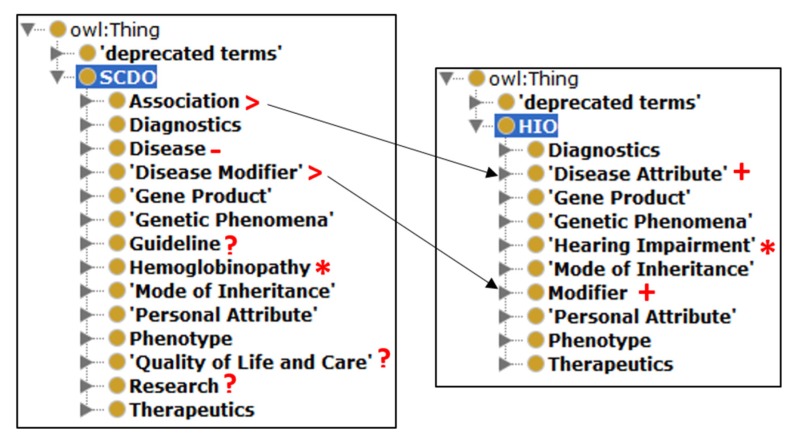
The upper classes of the Sickle Cell Disease Ontology (SCDO) and HI Ontology (HIO). (*) indicates the ontologies’ central classes. (+) indicates classes in HIO but not in the SCDO. (-) indicates a class in SCDO but not in HIO. (>) indicates classes in SCDO that were incorporated in other HIO classes. (?) indicates SCDO classes that still need to be reviewed and adapted as necessary for inclusion into the HIO.

**Figure 3 genes-10-00960-f003:**
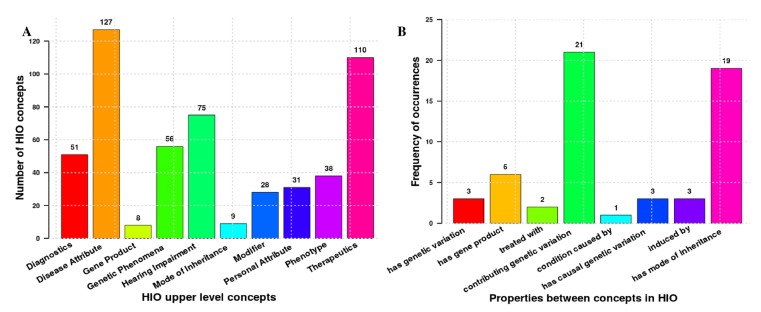
Summary statistics of current concepts and properties in the current HIO. Numbers at the top of bars represent the number of different HIO sub-classes topologically linked to upper level classes (**A**) and the occurrence frequency of a given property or association in the ontology (**B**). Note that ‘contributing genetic variation’ is used as the short hand label for the ‘gene carrying contributing genetic variation’ property and ‘has causal genetic variation’ for the ‘condition has causal or contributing genetic variation’ property.

**Figure 4 genes-10-00960-f004:**
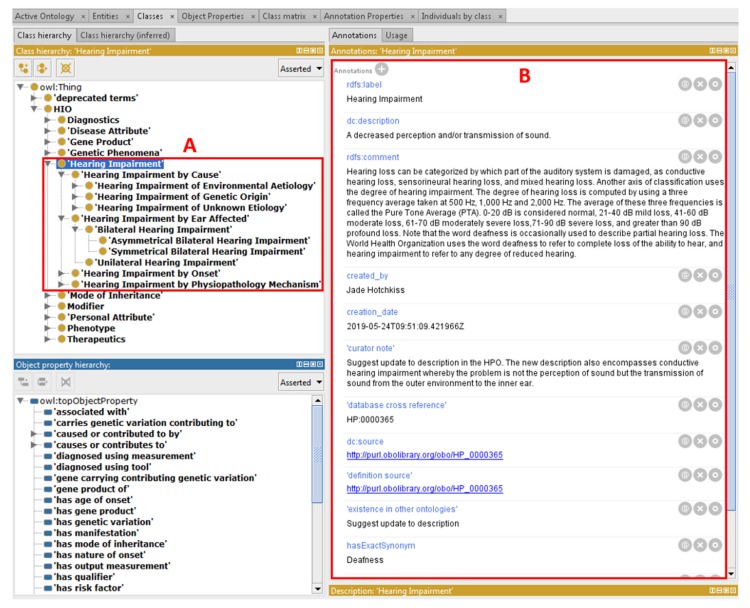
The ‘Hearing Impairment’ class within the HIO. (**A**) General categorization of hearing impairments in the ‘Hearing Impairment’ upper class and (**B**) annotations of the ‘Hearing Impairment’ class.

**Figure 5 genes-10-00960-f005:**
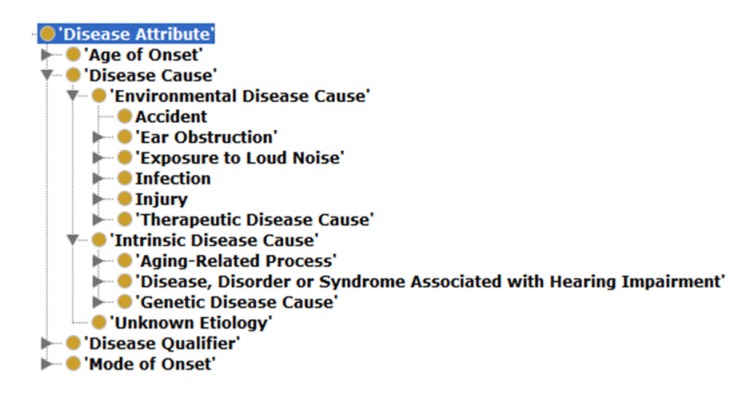
The ‘Disease Attribute’ class structure within the HIO. This hierarchy is intended to contain all possible features specific to or leading to HI.

**Figure 6 genes-10-00960-f006:**
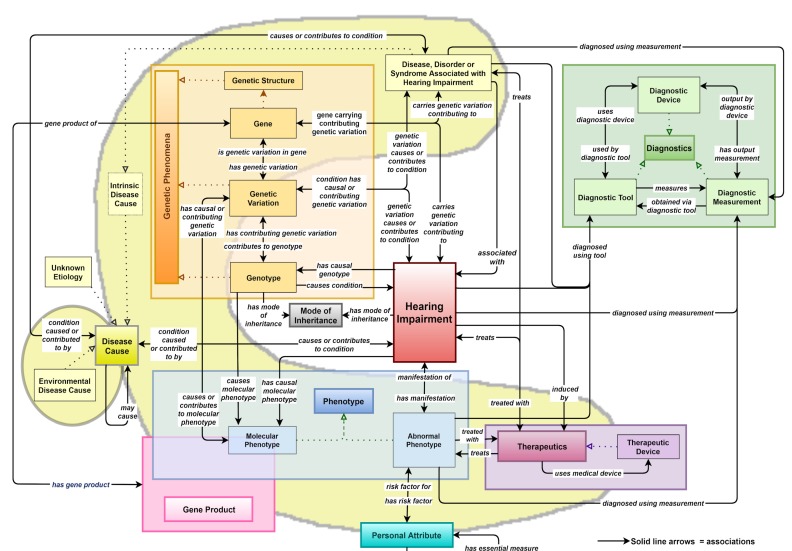
Associations made in the HIO between upper level (close to the root of the ontology) classes (excluding ‘Modifier’ class and only including ‘Disease Cause’ sub-class (yellow shapes) of the ‘Disease Attribute’ class). The ‘Hearing Impairment’ class is the central class.

**Figure 7 genes-10-00960-f007:**
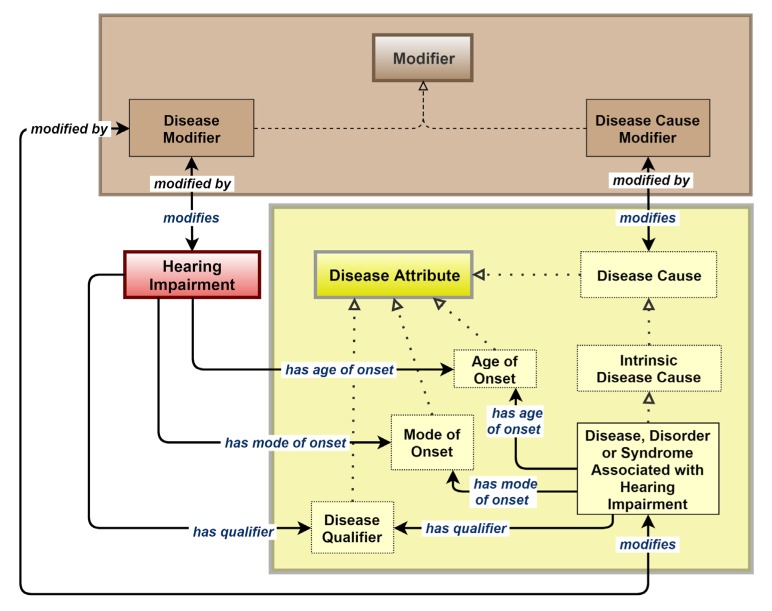
Associations made in the HIO to and from the ‘Modifier’ and ‘Disease Attribute’ classes, excluding those already shown in Figure 5 for the ‘Disease Cause’ class.

**Table 1 genes-10-00960-t001:** Summary of terms’ existence statuses prior to inclusion in the HIO.

Existence Status	Explanation of Status	No. Terms	% Terms
Sufficient	Exists in other ontology and has appropriate description	284	71.2
Suggest update to description	Used term from existing ontology but will suggest they update their description to ours	27	6.8
Suggest update to label	Used term from existing ontology but will suggest they update their label to ours	0	0
Suggest update to label and description	Used term from existing ontology but will suggest they update their label and description to ours	0	0
Few but definitions not available	Term exists in a few ontologies but has not been given a description in any	3	0.8
Few but definitions not freely available	Term exists in a few ontologies but the description is not freely available	8	2.0
Few but definitions not specific enough	Term exists in a few ontologies but the definitions are not specific enough for the HIO’s needs	9	2.3
Not relevant to context of hearing impairment	Term exists in other ontologies but the definitions are not relevant to the HI field	3	0.8
Negligible	No description or outdated ontology	2	0.5
None	Not in any existing ontology	63	15.8

**Table 2 genes-10-00960-t002:** Some of terms that are unique to HIO.

Term Label	Term ID	Term Description
Symmetrical Bilateral Hearing Impairment	HIO:0000365	When the severity and configuration of hearing impairment is approximately the same in both ears.
Asymmetrical Bilateral Hearing Impairment	HIO:0000366	When each ear has a different severity and configuration of hearing impairment.
Postlingual Hearing Impairment	HIO:0000475	Hearing impairment which develops after the acquisition of speech and language, usually after the age of six.
Prelingual Hearing Impairment	HIO:0000476	Hearing impairment which is either congenital or develops before the acquisition of speech and language, usually before the age of 6.
Temporal Bone Fracture with Otic Capsule Involvement	HIO:0000287	Traumatic injury to the temporal bone in which the continuity of the bone is broken and violation of the otic capsule is involved.
Temporal Bone Fracture without Otic Capsule Involvement	HIO:0000288	Traumatic injury to the temporal bone in which the continuity of the bone is broken and violation of the otic capsule is not involved.
Pseudo-Dominant Inheritance	HIO:0000228	When the inheritance of a recessive trait mimics a dominant pattern of inheritance.
Cisplatin-Induced Hearing Impairment	HIO:0000215	Hearing loss caused by cisplatin (a chemotherapeutic agent) ototoxicity.
Neomycin-Induced Hearing Impairment	HIO:0000285	Partial or complete loss of hearing following ingestion of neomycin.
Maternal Medical History	HIO:0000362	A record of a patient’s biological mother’s background regarding health and the occurrence of disease events of the mother.
Hearing Impairment based on Immaturity	HIO:0000514	Hearing impairment that occurs due to premature birth (birth at or before 37 weeks of gestational age).

**Table 3 genes-10-00960-t003:** Some existing online hearing impairment resources.

Scheme	Description	Types	URL	Reference
HHL	Hereditary Hearing Loss Homepage	An up-to-date overview of the genetics of hereditary hearing impairment for researchers and clinicians working in the field.	https://hereditaryhearingloss.org/	-
SHIELD	The Shared Harvard Inner Ear Laboratory Database	An integrative gene expression database for inner ear research	https://shield.hms.harvard.edu	[27]
DVD	Deafness Variation Database	A comprehensive resource integrating available genetic, genomic, and clinical data together with expert curation to generate a single classification for each variant in 152 genes implicated in syndromic and non-syndromic deafness.	http://deafnessvariationdatabase.org/	[28]
LOVD	Leiden Open Variation Databse	Retinal and hearing impairment genetic variant database	https://databases.lovd.nl/shared/genes/OTOF	[29]
NIDCD	National Institute on Deafness and Other Communication Disorders	A resource providing knowledge about Hearing, Ear Infections, and Deafness Diseases and Conditions. It also provides NIDCD Temporal Bone Registry at https://www.tbregistry.org/, a resource for learning about the pathology and pathophysiology of otologic disorders, which serves as a resource for scientists to analyze data from a collection of more than 12,000 temporal bone specimens.	https://www.nidcd.nih.gov/health/hearing-ear-infections-deafness	-
gEAR	Gene Expression Analysis Resource	Visualization and analysis of multiomic data both in public and private domains.	https://umgear.org/	-
OMIM	Online Mendelian Inheritance in Man	An Online Catalog of Human Genes and Genetic Disorders	https://www.omim.org	[30]

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
