# Peer review of "The Hearing Impairment Ontology: A Tool for Unifying Hearing Impairment Knowledge to Enhance Collaborative Research"

_genes, 2019, doi:10.3390/genes10120960_

Round 1
Reviewer 1 Report
The manuscript Hotchkiss and co-authors is very nicely crafted. If the Hearing Impairment Ontology becomes a reality, such a comprehensive resource of knowledge about human hearing loss will be unique and of considerable value. What is missing from the manuscript is a list of individual resources that are already available and widely utilized such as the Online Mendelian Inheritance in Man (OMIM) (https://www.omim.org), the Hereditary Hearing Loss Homepage, gEAR, SHIELD etc. There are many such sources of relevant information that be included in a table.
On page 10 of 12, the authors indicate that there is a dedicated curation team, which will require long term funding if the HIO is to be kept up to date. Note that the OMIM had long term dedicated funding until one day on very short notice, funding was terminated.
Author Response
Response to Reviewer 1 Comments
Point 1: The manuscript Hotchkiss and co-authors is very nicely crafted. If the Hearing Impairment Ontology becomes a reality, such a comprehensive resource of knowledge about human hearing loss will be unique and of considerable value. What is missing from the manuscript is a list of individual resources that are already available and widely utilized such as the Online Mendelian Inheritance in Man (OMIM) (https://www.omim.org), the Hereditary Hearing Loss Homepage, gEAR, SHIELD etc. There are many such sources of relevant information that be included in a table.
Response: Thank you to the reviewer for his appreciation and suggestion. As suggested, we have now included a table in the revised version of the paper, listing some of online hearing impairment resources (refer to Table 3, on Page 20) for illustration.
Point 2: On page 10 of 12, the authors indicate that there is a dedicated curation team, which will require long term funding if the HIO is to be kept up to date. Note that the OMIM had long term dedicated funding until one day on very short notice, funding was terminated.
Response: We thank the reviewer for this pertinent observation. So, as we were already aware of this, we used a model that consists of selecting volunteers from the current HIO team and experts in the HI field to keep updating the HIO. A similar model was used to develop for the Sickle Cell Disease Ontology, and it is still active to date. So, we believe that lack of funding would not be a serious constraint, in as much that we have dedicated volunteers and it is expected that the HIO team will expand and grow.
Reviewer 2 Report
The authors refer in this manuscript to the Hearing Impairment Ontology that they have created and that is available online. The idea of a comprehensive ontology seems interesting, although its applicability can be discussed. I do have some specific comments:
1) The Hearing Impairment Ontology (HIO) working group is described as a group with hearing impairment experts. Who are these experts? I believe these should be mentioned. Moreover, I do not see an ENT specialist, otologist or audiologist in the author’s list, which seems at least remarkable for a hearing impairment nomenclature. Apart from one of the senior authors, almost none of the authors have a publication about hearing loss in PubMed.
2) The methods to come to the definite ontology are not clearly described. Large concepts are mentioned in the Methods section, but it should be more specific. E.g. which terms were no longer accepted (synonyms), who decided on it and on which data was this based? Who made part of the expert group validating the concepts in the ontology? Was the two times checking of terms enough to be validated? Was there some kind of Delphi process in order to decide on different terms? On which base was decided which genes were included? Which syndromes?
3) Results, ontology:
- Where would you classify hearing loss based on immaturity (e.g. due to prematurity)?
- Why did you not elaborate 'Diagnostics'? I can imagine that inclusion of all diagnostic procedures could be interesting (if not necessary).
- Where would you classify acquired external auditory canal atresia, which is not included? Why is excessive cerumen in ear canal mentioned under 'intrinsic disease cause' and under 'environmental disease cause', which are on the same line?
- Where is CMV infection, which is one of the leading causes for HI in western world? Why is it not included, while rubella infection is? Same for meningitis as a cause for hearing loss. Where is otosclerosis as cause for hearing loss.
- The sublevels of therapeutics are all on the same level, which seems not correct. Curative therapeutics cannot be on the same level as drugs, as drugs are a treatment and curative is the intent of treatment.
Based on the above comments about the content, I hope you understand that the proposed ontology is largely incomplete and mentioning the fact that it is ongoing work in the discussion is not enough to overcome this.
Author Response
Response to Reviewer 2 Comments:
Point 1: The Hearing Impairment Ontology (HIO) working group is described as a group with hearing impairment experts. Who are these experts? I believe these should be mentioned. Moreover, I do not see an ENT specialist, otologist or audiologist in the author’s list, which seems at least remarkable for a hearing impairment nomenclature. Apart from one of the senior authors, almost none of the authors have a publication about hearing loss in PubMed.
Response: We thank the reviewer for this comment. It is known that an ontology is a multi-disciplinary application, bringing together scientists working in the different aspects of the domain being conceptualized in collaboration with ontologists. This initial HIO team is effectively constituted of researchers in hearing impairment (HI), as well as in biological ontology, in order to set this HIO core ontology. These researchers have been working in HI and biological ontology for years, each of them having a verifiable publication track in either HI or biological ontology. It is worth pointing out that some of these HI researchers are clinicians. Finally, This is just the beginning of a big HIO consortium we are dreaming of and which, we believe, will expand and bring other researchers, ontologists and specialists in different fields.
Point 2: The methods to come to the definite ontology are not clearly described. Large concepts are mentioned in the Methods section, but it should be more specific. E.g. which terms were no longer accepted (synonyms), who decided on it and on which data was this based? Who made part of the expert group validating the concepts in the ontology? Was the two times checking of terms enough to be validated? Was there some kind of Delphi process in order to decide on different terms? On which base was decided which genes were included? Which syndromes?
Response: The methods have now been clarified by including in the revised version of the paper a flow chart highlighting the methods described in the manuscript (see Figure 1 on Page 3). As indicated in the subsection “Distributed model-based HIO design” of the “methods” section (see Page 4), we used the online collaborative ontology development tool, WebProtege, which provides a highly distributed ontology content management system, enabling domain experts, ontology curators and developers to share and update information, and easily visualise the ontology classes and structure. After, there was a general online discussion to share a common understanding of existing HI knowledge currently included in the ontology. if there is any disagreement, it is then resolved during this general online discussion. This has now been clarified in the revised version of the paper (see subsection “HIO file refinement and evolution” in the “Methods” section, on Page 4, line 147).
Point 3: Results, ontology:
- Where would you classify hearing loss based on immaturity (e.g. due to prematurity)?
Response: The concept has now been included in the updated version of the ontology under the label “Hearing Impairment based on Immaturity” with synonym “Hearing Impairment due to Prematurity”, classified under ‘Neonatal Onset Hearing Impairment’ and ‘Hearing Impairment of Environmental etiology’.
- Why did you not elaborate 'Diagnostics'? I can imagine that inclusion of all diagnostic procedures could be interesting (if not necessary).
Response: Thank you to the reviewer for pointing out these omissions. Indeed, owing to the fact that an ontology is never complete and dynamic in nature as knowledge changes and increases over time, we acknowledge that the ‘Diagnostics’ class is not complete. This class has now been updated in the revised version of the ontology to include the most popular diagnostic methods. The number of concepts in this class has now grown from 8 initially to 51 and we intend to keep updating with the latest diagnostic methods.
- Where would you classify acquired external auditory canal atresia, which is not included? Why is excessive cerumen in ear canal mentioned under 'intrinsic disease cause' and under 'environmental disease cause', which are on the same line?
Response: Thank you to the reviewer for pointing out the omission of ‘external auditory canal atresia’. It has now been included in the revised version of the ontology, below ‘Abnormal External Auditory Canal Morphology’ (under ‘Intrinsic Disease Cause’). The classification of “Excessive Cerumen in Ear Canal” has now been fixed in the revised version of the ontology. It has been moved from below “External Ear Canal Obstruction” and set under “Abnormal External Auditory Canal Morphology” (under ‘Intrinsic Disease Cause’), according to its placement in the Mammalian Phenotype Ontology.
- Where is CMV infection, which is one of the leading causes for HI in western world? Why is it not included, while rubella infection is? Same for meningitis as a cause for hearing loss. Where is otosclerosis as cause for hearing loss.
Response: Thank you to the reviewer for pointing out these omissions. These terms, as well as others, e.g., Cytomegalovirus (CMV) Infection, Meningitis, Otosclerosis, Mumps, etc. have now been included in the revised version of the ontology.
- The sublevels of therapeutics are all on the same level, which seems not correct. Curative therapeutics cannot be on the same level as drugs, as drugs are a treatment and curative is the intent of treatment.
Response: Thank you to the reviewer for this observation. We have edited the description of Therapeutics in the updated version of the HIO, so that “Drug” and “Therapeutic Device” are appropriate as sub-classes of the different types of therapies.
Point 4: Based on the above comments about the content, I hope you understand that the proposed ontology is largely incomplete and mentioning the fact that it is ongoing work in the discussion is not enough to overcome this.
Response: Thank you to the reviewer for this comment. However, as pointed out previously, an ontology is neither complete nor static, and this is because an ontology is a representation of a universal knowledge of a specific domain. Thus, even long time known biological ontologies, e.g., “gene ontology”, “human phenotype ontology”, are never complete and permanently undergo 'update' and 'term addition' processes as new knowledge and discoveries come up. This is even more acute in this 'big data' age with high throughput technology generating large scale datasets from which knowledge can be extracted. It is expected that HI knowledge will continue to increase and to be updated, and that is why we have stated in the ‘Challenges and future direction’ subsection in the “Discussion” that ‘there is admittedly still much content to be included, both with regards to terms and associations’. Furthermore, we provided an update mechanism of the new HIO release (see subsection “HIO release and license” of the “Results” section).
Round 2
Reviewer 2 Report
Thank you for the modifications implemented in the manuscript. I feel the methods are more clear now and some required items have been added. Based on my previous suggestions, I still have the impression that the ontology should benefit from a more clinical point of view, but I can accept that it is work in progress and it is a decent fundament to start from.